# Systematic framework to assess social impacts of sharing platforms: Synthesising literature and stakeholder perspectives to arrive at a framework and practice-oriented tool

**Steven Kane Curtis**[1¤a]*, **Jagdeep Singh**[1], **Oksana Mont**[1], **Alexandra Kessler**[2¤b]

1 International Institute for Industrial Environmental Economics (IIIEE), Lund University, Lund, Sweden,
2 Collaborating Centre on Sustainable Consumption and Production (CSCP), Wuppertal, Germany

¤a Current address: IIIEE at Lund University, Lund, Sweden
¤b Current address: Collaborating Centre on Sustainable Consumption and Production, Wuppertal, Germany
* steven.curtis@iiiee.lu.se

**Data Availability Statement:** All relevant data are within the manuscript and its Supporting Information files.

## Abstract

### (1) Background

Research and user experience suggests both positive and negative social impacts resulting from practices in the sharing economy: social cohesion vs. gentrification; inclusiveness vs. discrimination; flexible employment vs. exploitation. However, as yet, there is no framework for understanding or assessing these social impacts holistically.

### (2) Objective

We aim to improve understanding of the social impacts of sharing platforms and develop a systematic framework to assess these impacts.

### (3) Methods

We conduct a narrative literature review and stakeholder workshop, integrating insights to produce a systematic social impact assessment framework and a practice-oriented tool.

### (4) Results

We identify four social aspects—trust, empowerment, social justice, and inclusivity—and eighteen indicators that make up the framework. We describe each indicator and its relevance to the sharing economy as well as suggest measurable variables in the form of a practice-oriented tool.

### (5) Conclusions

The framework and tool are the first holistic method for assessing social impact in the sharing economy, which may inform researchers, sharing platforms, regulators, investors, and

**Funding:** This research has received funding (OM) from the European Research Council (ERC) under the European Union's Horizon 2020 research and innovation programme (Grant Agreement No. 771872). This research was also funded by the German Federal Ministry for Education and Research (BMBF) (Grant Agreement No. 01UU1701C) in the context of the project Urban Up – Upscaling Strategies for an Urban Sharing Society. No commercial companies funded this research, nor did any authors receive salary or other funding from commercial companies. The funders had no role in study design, data collection and analysis, decision to publish, or preparation of the manuscript. https://erc.europa.eu/ https://www.bmbf.de/en

**Competing interests:** The authors have declared that no competing interests exist.

citizens to mitigate adverse social impacts while enhancing the overall net social value of the sharing economy.

## Introduction

The sharing economy is said to embody values of openness, trust, empowerment, and a sense of collectivism. Proponents claim that the sharing economy empowers people, creates trust among strangers, builds social capital, and promotes social cohesion [1]. However, there is a spectrum of consumption practices ascribed to the sharing economy that generate conflicting social impacts, e.g. "true sharing" vs. pseudo-sharing [2, 3]. "True sharing" refers to ". . .distributing what is ours to others for their use and/or the act and process of receiving or taking something from others for our use" [4]. Consumption practices most resembling "true sharing" are better suited to facilitating communal links and socialisation [3], but most practices attributed to the sharing economy are not considered sharing at all [5]. Instead, other consumption practices are often conflated under the term "sharing economy", such as renting, leasing, borrowing, lending, bartering, swapping, trading, exchanging, gifting, buying second-hand, and even buying new goods [6]. In addition, practices such as time banking [7], collaborative production (e.g. makerspaces) [8], and the gig economy [9] are also often included under the banner of the sharing economy.

Research and popular media promote practices attributed to the sharing economy, claiming these consumption practices have the potential to facilitate more open, inclusive, and democratic modes of production and consumption [10]. However, the promise of the sharing economy contrasts with the practices of sharing platforms, which lead to varying experiences among users and society: inclusiveness vs. discrimination, democratisation vs. social exclusion, flexible employment vs. exploitation, social cohesion vs. gentrification. Growing empirical evidence and user experiences suggest negative social impacts as a result of the activities of sharing platforms, e.g. Airbnb [11–13], Uber [14–16], and other shared mobility platforms [17]. There is concern that sharing platforms are simply exploiting time and resources of their users to their detriment [18].

While these paradoxes have been explored by others [8], there is recognition of the growing need to assess the social impacts of sharing platforms [10]. However, knowledge about social impacts in the sharing economy remains scarce and fragmented [19]. Most studies tend to advocate specific perspectives, for example, trust [20, 21] and discrimination [22, 23], while other studies focus on discussing a range of social impacts emanating from a single sharing platform such as Airbnb [11] and Uber [14]. This narrow focus results in relatively limited conceptual transferability across the diversity of sharing economy business models and related consumption practices. A more holistic framework to assess social impacts of sharing platforms would advance research on the sharing economy, support sharing platforms to understand/prioritise their social impacts, and inform policymakers interested in safeguarding consumer safety, while promoting the potential societal benefits of the sharing economy.

Assessing social impacts from sharing platforms is difficult. Methods, conceptual frameworks, or practical tools are lacking [24]; assessing perceived social impacts is often qualitative and requires value judgements or prioritisation that may be uncomfortable [25]; and, sharing platforms may be unwilling to collect or share data or may even lack the data (e.g. limited resources or access to users).

The aim of this research is three-fold: 1) to improve understanding of the social impacts of sharing platforms; 2) to develop a systematic framework to assess the social impact of sharing platforms; and 3) to operationalise the framework by proposing a practice-oriented tool that will allow sharing platforms to self-assess their social impact, as well as inform other interested stakeholders. We do so by building our framework around four broad social aspects: trust, empowerment, inclusivity, and social justice.

The article proceeds by offering background literature introducing the sharing economy, discussing social impacts of sharing platforms, and providing an overview of existing tools to assess social impacts. Then, we describe methods employed in this study, including literature review, stakeholder workshop, and development of our framework. We review the findings from a stakeholder workshop in Sweden, and consolidate these findings with literature to develop a systematic framework. A preliminary tool is proposed before we discuss key findings, contributions, and conclusions as well as outline possibilities for future research.

## Background literature

The potential of the sharing economy to contribute towards social sustainability has been the focus of much debate [8, 26, 27]. However, it is difficult to assess social impacts, which advance or hinder social sustainability. Furthermore, understanding of these concepts in academia remains contested. Stakeholders—including sharing platforms, managers, regulators, investors, and citizens—are interested in understanding the social impacts of sharing platforms, and need clear methods and tools, which overcomes the fuzziness of concepts presented in academic literature. Therefore, we strive to strike this balance and elaborate on our understanding of the sharing economy, the social impacts resulting from its diverse consumption practices, and its contribution to social sustainability.

### The sharing economy

Broadly, the sharing economy is said to facilitate access over ownership by making use of the idling capacity of goods and services, often leveraging technology to improve the economic efficiency of sharing [28]. It's growth in the last decade is described as a response to the 2008 financial crisis and a malfunctioning global financial system [29], where the sharing economy enables citizens to maintain a decent standard of living [30] through greater access to goods and services [31]. At the same time, advancements in information and communication technology (ICT) have reduced transaction costs associated with sharing among strangers, leading to increased levels of supply and demand and platforms benefiting from economies of scale. Transaction costs are understood by economists as the total costs (monetary and non-monetary) associated with making any economic transaction, including time and resources needed to access the market, to facilitate suitable offers, and to organise contracts or transactions [32]. Platforms in the sharing economy rely on technology and algorithms to match users, facilitate ratings and reviews, process payments, among other activities, thereby increasing the extent and ease of information exchange between the involved parties [33].

On the basis of a systematic literature review [28], we define the sharing economy as "...a socio-economic system that leverages technology to mediate two-sided markets, which facilitate temporary access to goods that are under-utilised, tangible, and rivalrous" [6]. We consider the system of actors involved in the sharing economy to include sharing platforms, their users, and society. We use the term user to include the actors involved in the two-sided market. The actor on the supply-side of the market, we call the resource owner; the actor on the demand-side of the market, we call the resource user [6]. We define society to include citizens

broadly as well as municipal representatives, media, academia, civil society, business associations, and other interested actors.

We focus our attention on sharing platforms as the mediator of the consumption practice, which facilitates social value creation. Sharing platforms connect a resource owner (the platform user providing access to a good they own) to a resource user (the platform user accessing a good that someone else owns) in order to facilitate access to under-utilised goods [6]. This emphasises sharing as a practice, with various impacts (both positive and negative) for the platform, its users, and society.

Yet, Belk [2, 4, 34] suggests that sharing is different from other consumption practices that extend product lifetime (gifting, second-hand, commodity exchange) because there is not necessarily the need for reciprocity or compensation. As such, depending on the user's motivation, interactions may range from transactional (e.g. renting) to prosocial (e.g. sharing), indicating that the motivation among users affects the outcome (i.e. social impact) of sharing via a platform. Bucher, Fieseler & Lutz [35] demonstrate that non-commercial users are more likely to hold moral and social-hedonic motives, unlike commercial users. Sharing platforms that operate as cooperatives, i.e. platforms owned and operated by its users, are more likely to realise positive social impact [36, 37]. Therefore, the design of the sharing platforms, user motivations, and subsequent consumption practices are important when considering the social impacts resulting from sharing [38]. It is from this point of departure that we develop a framework and practice-oriented tool to assess the social impact of sharing platforms.

## Social impacts of sharing platforms

The social impacts of sharing platforms are diverse and complex, subject to differing understandings and priorities based on the actors involved in sharing [38]. For example, platforms may advantage some users while disadvantaging others. Meanwhile, the actions of users also impact each other as well as the platform and its ability to continue to provide services for others and broader societal impact. Finally, society at large is impacted when some groups are included or excluded, exploited for their labour or resources, or gentrified. We seek to capture this complexity in developing our work around four social aspects—trust, empowerment, inclusivity, and social justice (Table 1)—which capture many of the positive and negative social impacts discussed in literature already attributed to the sharing economy.

In relation to these aspects, sharing platforms are said to facilitate positive impacts such as enhancing social cohesion, increasing trust in communities, empowering individuals, and increasing social ties among strangers [8]. Trust has been identified as one of the most critical issues that serves as a "lubricant" in the sharing economy [39, 40]. Participation in sharing platforms may increase trust in peers [41, 42], in platforms [43], and in technology [44, 45].

**Table 1. Definitions of the social aspects considered in this study.**

| Social Aspect | Definition | Relevant Literature |
|---|---|---|
| Trust | Trust is the assured reliance on the character, ability, strength, or honesty of someone or something. | Hawlitschek et al. (2016, 2018); Huurne et al. (2017); Mazzella et al. 2016); Parigi & Cook (2015) |
| Empowerment | Empowerment is the action of enabling someone or something, by granting power, privilege, or authority as well as providing the necessary support, communication, or resources to motivate and inspire. | Füller et al. (2009); Mäkinen (2016); Pires, Stanton & Rita (2006) |
| Inclusivity | Inclusivity is the quality of trying to involve many different groups of people in decision-making and governance, emphasising the need for broader consultation and engagement of diverse communities, particularly those vulnerable or marginalised. | Ferrari (2016); George, McGahan & Prabhu (2012); Oxoby (2009) |
| Social Justice | Social justice is the quality of being equitable, impartial, or fair, including the *distribution* of benefits, the *representation* of diverse groups, and the *participation* of those groups. | Cribb & Gewirt (2003); Eubanks (2012); Gardner, Holmes & Leitch (2009) |

Studies have examined the role of platforms and their practices to foster trust online [46], and others have focused on the dichotomy between social inclusion and exclusion [47–49]. Research suggests that sharing platforms provide access to goods and services for people who could otherwise not afford them, as well as the possibility to generate extra income from one's own goods or skills, which has been framed as increasing inclusion, empowerment, and justice [49–51]. By connecting with strangers, sharing platforms are argued to increase social interaction between people, fostering social cohesion within local and global communities [52]. This may also lead to empowerment of marginalised social groups, e.g. women [49]. Some studies identify improvement of urban space through sharing, such as revitalisation of space, reduced pollution, better connectivity [53–55], as well as cultivating conscious tourists and communities due to alternative forms of consumption [56, 57].

Recent attention is also turning to the negative impacts of sharing platforms, for example, discrimination [22, 23], gentrification [58], casualisation of labour [59], and commodification of relationships [60]. The latest studies on accommodation sharing draw attention to over-tourism, touristification, and tourism-phobia in cities where conflicts are growing between tourists and the local population [61–65]. More specifically, the following negative impacts have been identified for Airbnb: non-civic behaviour, such as noise, vandalism, and violence [66, 67]; crowding-out of the long-term rental market, resulting in conflicts between resident Airbnb hosts and non-hosts [11]; increasing housing prices driven by short-term rentals [19, 68]; and overcrowding of cities with mass tourists [62]. As such, gentrification has been identified as a negative impact of accommodation sharing [58].

Other negative impacts include the exclusion of social groups, e.g. the poor or elderly, who may not possess the requisite technology or skills to participate [49]. Studies demonstrate that increased social interaction between strangers may also result in various forms of discrimination [22, 69–71]. Concern is also growing over data protection and personal privacy among users of sharing platforms [72]. This is to say nothing about the gig economy, occasionally included under the banner of the sharing economy, where work is often unstable, informal in nature, and lacking access to organised labour unions. This often leads to precarious work situations, without typical work-related securities, benefits, or similar [73, 74].

Whether positive or negative impacts, the distribution of these impacts is not experienced equally. Early adopters of sharing platforms tend to be younger, educated, more affluent, and more socially-connected [57, 75, 76]. Sharing platforms can theoretically support lower layers of society, e.g. democratising consumption and providing greater access to resources otherwise unattainable. However, there is not yet sufficient evidence that this happens widely, due to inequities of time, resources, and access to the internet. Additionally, research demonstrates that sharing platforms adversely affect incumbents [77] and municipalities [78], specifically around issues including competition, consumer safety, casualisation of labour, and tax avoidance [79, 80].

### Social sustainability and social impact assessments

Much like the sharing economy, social sustainability is a concept taken for granted by numerous disciplines [81]. Social sustainability is described as a multi-dimensional concept focusing on the shared social goals of sustainable development [81, 82]. These goals often relate to personal well-being as well as meaningful interactions with others [83]. While there is no consensus on specific outcomes, literature does provide overlapping concepts relevant to social sustainability, for example, social capital, social cohesion, social inclusion, and social justice [81]. Due to the fluidity of concepts and the challenges associated with prioritising outcomes

contributing to social sustainability, we take inspiration from social impact assessments (SIAs), as elaborated by Assefa & Frostell et al. [83].

SIA is "...the processes of analysing, monitoring, and managing intended and unintended social consequences", which includes both positive and negative impacts resulting from a focal intervention [81]. Within the field of SIA, these social impacts describe changes to a person or people's way of life, culture, community, political system, environment, health and wellbeing, personal and private property rights, as well as fear and aspirations [83, 84]. In this study, we depart from this understanding to assess the social impacts of sharing platforms as a proxy for contributing to social sustainability.

A number of tools have been developed to assess social impact broadly. For example, the International Guidelines and Principles for Social Impact Assessment describe processes for evaluation of the intended and unintended social consequences of policies, programmes, plans, and projects [84]. The International Standards Organization (ISO) has developed the standard ISO 26000 that provides guidance on social responsibility [85]. The standard classifies social aspects into seven themes—human rights, labour practices, the environment, fair operating practices, consumer issues, community involvement, and development [85]. In recent decades, increased effort has focused on processes to develop social indicators [86] and procedures to measure social impact [87]. Social impact is becoming an integral part of sustainability assessments among global organisations such as the ISO [85], United Nations Global Compact [88], Global Reporting Initiative [89], and the Organisation for Economic Co-operation and Development [90].

Across these approaches, there are numerous ways to classify social impacts among a range of categories, comprising qualitative indicators, classifications, and assessment criteria as well as some quantitative indicators. They have usually been developed for organisations or activities with relatively formal structures, while sharing platforms also comprise informal initiatives, activities, and networks. In contrast, research on assessing the social impacts of sharing platforms tends to focus on specific social impacts, e.g. trust [20, 21] and discrimination [22, 23], or may address social impact from one particular practice, e.g. food sharing [24]. However, this often results in a limited number of considered social impacts and relevant indicators, leaving many social impacts unaccounted. for, so are less beyond food sharing.

To date, there is no systematic framework to assess the social impact suitable for the wide diversity of sharing platforms and practices, with the exception of a recent study by Laukkanen & Tura [91]. They developed a general framework that classifies social aspects of sharing economy business models into five categories: safeguarding health and safety; respecting laws, regulations, and rights; respecting employee, stakeholder and individual rights; ethical principles; and no harmful social impacts and increasing social well-being [91]. The framework is rather general, which could be improved by: 1) increasing the level of granularity and decomposition of social impacts; and 2) taking into account the perspective of the main actors involved in the practice of sharing—platforms, resource owners, resource users, and society. While some studies have explored stakeholders' views on social sustainability of sharing [11], assessment of social impact from their perspective has rarely been addressed.

While there is a large body of extant knowledge on assessing social impacts, this has not been tailored to sharing platforms, resulting in varied understandings of their impact as well as fractured approaches to assessing their social impact. It is clear that their social impacts vary across shared practices (e.g. space, mobility, goods, consumables, resources) as well as platform types (e.g. peer-to-peer, business-to-business, business-to-peer, and crowd/cooperative) [6]. There is a need to develop a systematic framework as well as tools for assessing social impacts of sharing platforms. There is also a need to elaborate specific measurable variables for each stakeholder group participating in and/or impacted by sharing platforms. Such a framework

must be context-specific to the sharing economy, while adaptable to shared practices, platform types, and the broad range of stakeholders involved in and affected by the sharing economy.

## Methodology

We draw on both literature and empirical data to develop the systematic framework and subsequent tool, applying a multi-step methodology (Fig 1). We conducted a preliminary literature review in order to understand the current discourse on social impacts of sharing platforms and business models more broadly (Step 1). A stakeholder workshop was then held, to gain an impression of the broad perspectives on social impact in the sharing economy (Step 2). The data from the stakeholder workshop was analysed and refined in a series of workshops by the authors, complemented by a subsequent narrative literature review of social sustainability impacts of sharing platforms (Step 3). Based on the analysed data, a social sustainability assessment framework was developed (Step 4) and operationalised in the form of a practice-oriented tool (Step 5).

### Step one: Preliminary literature review

The purpose of the preliminary literature review was to get a broad overview of the social impacts of sharing platforms, which informed a discussion on social sustainability aspects and indicators during a stakeholder workshop. We describe *aspects* as social values that can be influenced by the sharing platforms and *indicators* as measurable criteria to approximate social

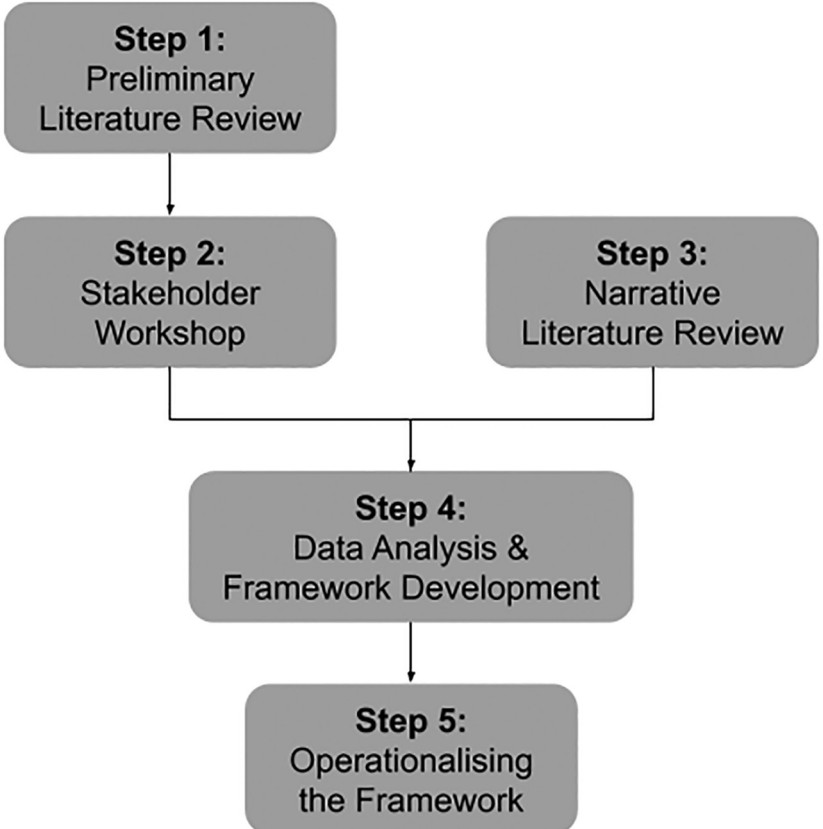

**Fig 1. Multi-step methodology to develop and operationalise framework.**

impact. Our preliminary review began in the autumn of 2018 with 36 relevant articles from the database of sharing economy literature, collated by Laurenti et al. [92]. The abstracts and keywords of these articles were analysed to identify common terms used to indicate, describe, or measure social impacts. Twenty-five keywords were identified and used in a subsequent online database search using Scopus, in combination with "sharing economy" OR "collaborative economy" search strings. This search resulted in 42 additional articles, and the same process of keyword identification was repeated. An additional 25 keywords were identified, resulting in a total of 50 keywords used to indicate, describe, or measure social impacts of the sharing economy (S1 Appendix).

These 50 keywords were structured thematically based on patterns in the data, resulting in several broad social sustainability aspects (S2 Appendix). However, some of these aspects were found to be interrelated and overlapping, i.e. the same social impact could be related to multiple social aspects. We merged overlapping concepts to arrive at four social aspects—empowerment, trust, inclusivity, and social justice. Based on the literature, we defined these aspects in order to (1) clarify their relevance for assessing social impacts of sharing platforms, and (2) identify the distinct and least overlapping social impacts. The results from our preliminary literature review informed our preparation for the stakeholder workshop.

## Step two: Stakeholder workshop

The stakeholder workshop took place at the Swedish National Laboratory on Sustainable Lifestyles in November 2018 in Kalmar, Sweden. The purpose of the workshop was to obtain feedback on the social aspects identified in the literature review and to co-create measurable indicators to assess social impacts in relation to each aspect. Thirty-five participants attended the workshop voluntarily during parallel sessions, based on interest. Previously, we had determined that our study did not meet the standard set by the Swedish Ethical Review Act (SFS 2008:192)–namely, Section 3 and Section 4, pertaining to applicability—requiring prior ethical approval of research involving humans. We did not collect or store any sensitive personal data, nor did we subject research participants to any physical intervention or risk of physical or mental injury. Data has been anonymised and aggregated, to avoid the identification of research participants.

Workshop participants were divided into 9 groups, based on their respective stakeholder categories. Five stakeholder categories were represented—companies, special interests, cities, public authorities, and academia. We broadly defined these stakeholder categories: 'companies' include sharing platforms, incumbents, and other formal or informal organisations; 'special interests' were industry associations, consumer organisations, and non-governmental organisations; 'cities' were individuals associated with any municipal government; 'public authorities' were national or regional agencies; and 'academia' was researchers or students. Participants were asked to consider both their role corresponding to their stakeholder category as well as a private citizen and/or user of sharing platforms.

The definitions of the four social aspects identified during the preliminary literature review were introduced to the participants and contextualised in relation to the sharing economy (S3 Appendix). An introductory explanation was given about what is an indicator and how it may be operationalised to measure social impact of sharing platforms. This brief introduction sought to clarify the aim of the workshop and stimulate discussions among the groups. Participants were then asked to collaborate within their groups on the following tasks:

1. Define these social aspects from the perspectives of their stakeholder group;

2. Suggest measurable indicators for each of the respective aspects.

Three workshop facilitators answered questions from participants regarding the tasks, but did not actively engage in the discussions. At the end of the workshop, participants were asked to rank the aspects by importance, balancing their stakeholder perspective and their perceived ease in measuring them. Then, participants were asked to record their final group responses in a worksheet that they submitted at the end of the workshop. Finally, the data was structured to summarise stakeholder insights relevant to the framework.

### Step three: Narrative literature review

Triangulated with the structured data from the stakeholder workshop, a subsequent narrative literature review [93, 94] supported our efforts to identify, describe, and operationalise social indicators. This approach is exploratory and less rigorous than a systematic literature review, but appropriate in this case when researchers had pre-existing knowledge in the subject area and literature on the subject was insufficient [94]. Our review also builds upon the preliminary literature review and the work by co-authors Kessler and Singh et al. [92].

The review was conducted in March 2020. Using the Scopus database, an initial search ["sharing economy" AND "social impacts"] was limited to title, abstract, and keywords, which resulted in 22 documents, comprising articles, conference papers, and book chapters. The abstracts were analysed, but none of the articles were found to be particularly relevant to developing a framework to assess social sustainability of sharing platforms. A second search ["sharing economy" AND "social impacts"] was expanded to consider the entire content of each document, resulting in 133 documents. The title, abstract, and keywords were reviewed and 41 documents were found relevant.

A complementary search identified literature that described or performed social sustainability assessment in the sharing economy using the query: TITLE-ABS-KEY ("sharing economy") AND ALL ("social sustainability") AND ALL (assessment OR evaluation OR tool OR framework OR indicators). This resulted in 35 documents that were reviewed, which added a further six documents to our final sample.

A final search used the query: ALL ("sharing economy") AND ALL ("social impact assessment") OR ALL ("social impacts") OR ALL ("social sustainability"). This produced 240 documents that were reviewed, resulting in 15 documents added to our final sample. Of those additional documents, only two suggested indicators or tools for measuring social impacts of the sharing economy: Laukkanen and Tura [91] and Mackenzie and Davies [24]. In total, 62 articles (41, 6, and 15 documents, from each search respectively) comprised the final sample from the database search.

### Step four: Data analysis and framework development

To develop our framework, we created a series of prototypes, which abductively incorporated literature and stakeholder feedback. The first prototype was informed by the preliminary literature review, which focused on four aspects of social sustainability. These four aspects were presented at the stakeholder workshop, from which the input was collected as data. This data was analysed using a constructivist grounded theory approach, which considers how and why participants construct meaning [95, 96]. This approach was chosen to consider the various stakeholder perspectives, which we mirror in our preliminary tool considering the sharing platform, resource owners, resource users, and society. Following the approach described by Kenny & Fourie [96], one researcher prepared the initial coding, seeking to develop relevant indicators and measurable variables. Two researchers then engaged in refocused coding in a series of three workshops, where the coding was refined and operationalised.

During these workshops, insights from literature also informed our coding. NVivo was used to support the subsequent coding of literature into categories, combined with data from stakeholders. Literature also informed our description of each indicator in relation to the sharing economy. The resulting analysis arrived at a final prototype of the framework to assess social sustainability of sharing platforms, considering a stakeholder perspective.

## Step five: Operationalising the framework

To operationalise our framework, we sought to validate and test the framework, resulting in a practice-oriented tool. First, we presented early prototypes of the framework to researchers and practitioners at the 6th International Workshop on the Sharing Economy (27–29 June, 2019) and the Nordic Sharing Cities Summit (10–11 October, 2019). Solicited feedback at these events informed the final prototype of the framework. We also collaborated with a sharing platform called FLOOW2, based in the Netherlands, to test and adapt our framework to their context. FLOOW2 works with clients to design and implement a sharing platform within a business or industry, for example, within hospitals, schools, and construction companies. We interviewed representatives from the platform and co-developed a user and citizen survey. Based on our collaboration and the exercise of adapting our framework to their context, we found that a more specific tool with measurable variables would be needed to support the assessment of social impacts.

Based on stakeholder perspectives, literature, feedback, and experience, we developed a practice-oriented tool to assess the social impact of sharing platforms. The tool provides potential measurable variables and sources of data for each indicator and all actors involved in or impacted by the sharing practice, e.g. sharing platform, resource owner, resource user, and society.

## Stakeholder insights

Data collected at the stakeholder workshop provides insights into the divergent perspectives among stakeholders regarding importance or priority of the various social aspects of sustainability. Five stakeholder categories—companies, special interests, cities, public authorities, and academia—were divided into nine groups during the workshop (Table 2). Their tasks were to define and/or expand the social aspects presented for discussion, propose measurable indicators, and rank the aspects based on their perceived level of importance.

First, the groups elaborated the descriptions for each social sustainability aspect—trust, empowerment, inclusivity, and social justice—based on their own experiences and perspectives. The mode of data collection allowed us to analyse the descriptions according to

**Table 2. Workshop groups and corresponding stakeholder categories.**

| Group | Stakeholder Category |
|---|---|
| Group 1 | Companies |
| Group 2 | Special Interests |
| Group 3 | Special Interests |
| Group 4 | Cities |
| Group 5 | Cities |
| Group 6 | Cities |
| Group 7 | Public Authorities |
| Group 8 | Public Authorities |
| Group 9 | Academia |

stakeholder category. Stakeholders suggest *trust* is something that must be earned and maintained, and that can be lost quickly in the face of scandal or lack of transparency (*Cities*). Transparency is an important factor in building trust (*Cities*), through compliance with standards and certifications (*Public Authorities*) and available data regarding positive and negative impacts (*Special Interests*). Trust is based on mutual integrity between users and the platform (*Company*; *Public Authorities*). Mutual trust is best achieved without direct interventions by the platform (*Special Interests*) and may lead to more robust levels of economic stability (*Public Authorities*) and improved trust in society, technology, and digital platforms (*Cities*).

Stakeholders described *empowerment* as a feeling of being a part of something bigger than oneself (*Company*), users feel they have a voice and sense of ownership (*Special Interest*) as well as the ability to influence the governance of the platform (*Special Interest*; *Cities*; *Academia*). Participation in processes of governance is an important aspect of empowerment (*Cities*; *Academia*), and platforms must be willing to share knowledge and skills (*Special Interest*). On a broader level, inclusive participation in a sharing platform can empower people to feel that they can shape the city and shift power from public and commercial interests to civil society (*Cities*). In this way, users, platforms, and society have changing roles and responsibilities for investment, maintenance, and disposal of shared resources (*Special Interests*).

Stakeholders described *social justice* as acknowledging the unfairness and inequitable structures in society by creating a safe space for different groups to participate in the platform based on their needs (*Cities*). Processes that prevent judgement, bias, and discrimination and respect privacy and personal data must be made fairer (*Companies*; *Special Interests*; *Public Authorities*). With greater availability, sharing should and can be part of creating a more equal society (*Public Authorities*).

Stakeholders related *inclusivity* to social justice. The groups described inclusivity as equal participation in decision-making (*Companies*), where everyone can join and share in the benefits of the platform (*Special Interests*; *Public Authority*). This requires platforms to actively reach out to all groups, regardless of whether they are using the service (*Cities*) including marginalised groups who do not normally feel included or involved (*Special Interests*). Accessibility is important, to foster inclusivity as well as friendliness and welcoming communication (*Cities*). Ideally, all people regardless of race, ethnicity, gender identity, sexual orientation, disability, religion, or age (*Cities*) should be welcome (*Public Authorities*), visible, and included in the activities of the platform (*Companies*).

Next, the workshop groups suggested measurable indicators based on their experience and priorities. The nine groups proposed 37 indicators across the four aspects presented and discussed at the workshop (Table 3). Stakeholders reported difficulty in proposing indicators due to the perceived challenges of measuring data or accessing data from platforms, their users, and society. While the indicators were suggested for specific social aspects, we see potential overlap that must be considered in the development of our assessment framework and tool. Trust had by far the highest number of proposed indicators (18), and social justice the lowest (4).

The number of indicators proposed by the workshop groups reflects the overall rank of social sustainability aspects. As a final task, workshop groups were asked to rank the aspects based on their experience and priorities as stakeholders (Table 4). Groups provided the same ranking for aspects when there was no clear priority among the members of the group. Due to time constraints, the final ranking was only received from 6 of the 9 groups. Trust was ranked highest (and had the highest number of proposed indicators) followed closely by empowerment. Social justice was ranked third with inclusivity being the lowest ranked social sustainability aspect. The task of ranking exemplifies how different stakeholders prioritise different

Table 3. Proposed indicators by stakeholders.

| Social Aspect | Proposed Indicators / Measurable Variables |
|---|---|
| Trust | • Presence of a review system<br>• Evaluation of the review systems by third party<br>• Number of reviews in relation to number of transactions<br>• Availability of users' information<br>• Transparency of platform communication (contact us, problem solving)<br>• Transparency in access to information (governance)<br>• Transparency about environmental, social, and economic impacts<br>• Measure extent to which rules are followed by users; codes of conduct<br>• Number of users/times resource is shared (as a proxy)<br>• Extent to which experiences/resources match the provided information<br>• Number of resources that are lost, disappeared, or broken<br>• Perceived safety over the platform<br>• Returning customers, platform reviews/reputation/complaints<br>• Presence of auditing system (national/international)<br>• Sharing data with local, regional, national governments<br>• Concrete measures that address problems seriously<br>• Degree of profit motive<br>• Customer satisfaction |
| Empowerment | • Amount of additional earnings<br>• Number of people using the service (as a proxy)<br>• Extent of participation/engagement in the platform (as a proxy)<br>• Acquired knowledge/skills<br>• Access to new forums and resources<br>• Perceived access/control/influence of platform, sense of ownership<br>• Number of platform initiatives fostering empowerment<br>• Demonstrable examples of how users contribute/are heard<br>• Type and extent of participation in governance<br>• Level of active participation in governance |
| Social Justice | • Reduced reliance on social support<br>• Representation of different socio-economic groups and under-represented groups in decision-making<br>• Mechanisms for sharing profit/benefits among the users<br>• Accessibility (e.g. language, contact us, flexible opening hours) |
| Inclusivity | • Inclusion of different socio-economic and under-represented groups of people in decision-making<br>• Possibility of citizens (i.e. non-users) to make suggestions or participate in dialogue<br>• Number of loans by e.g. age, race, gender, proportional to society<br>• Propensity to lend things to friends, neighbours, acquaintances<br>• Criteria/targets by platforms in how to include groups (e.g. targeted communication) |

aspects, and does not inform the significance or weight of any aspects or indicators presented in our framework.

## Social sustainability framework

Integrating literature and stakeholder perspectives collected in the workshop, we developed a framework to assess the social impact of sharing platforms (Fig 2). The framework presents eighteen indicators across four social aspects: trust, empowerment, social justice, and

**Table 4. Rank of social sustainability aspects by workshop groups.**

| Group # | Trust | Empowerment | Social Justice | Inclusivity |
| --- | --- | --- | --- | --- |
| G1 | 1 | 3 | 2 | 2 |
| G2 | 2 | 1 | 3 | 3 |
| G3 | 2 | 1 | 4 | 3 |
| G5 | 3 | 2 | 1 | 4 |
| G8 | 1 | 4 | 2 | 3 |
| G9 | 2 | 1 | 3 | 3 |
| Average | 1.8 | 2 | 2.5 | 3 |
| Final Rank* | **1** | **2** | **3** | **4** |

\* Final rank was based on the average, not the frequency, of ranks.

inclusivity. Each indicator seeks to assess the perceived experience from all stakeholders impacted by sharing as a practice including sharing platform, resource owner, resource user, and society.

In developing the framework, the indicators needed to remain abstract enough to be adapted to the context of the sharing platform and its particular sharing practice, but specific enough to be operationalised. We present the framework and describe each indicator, and

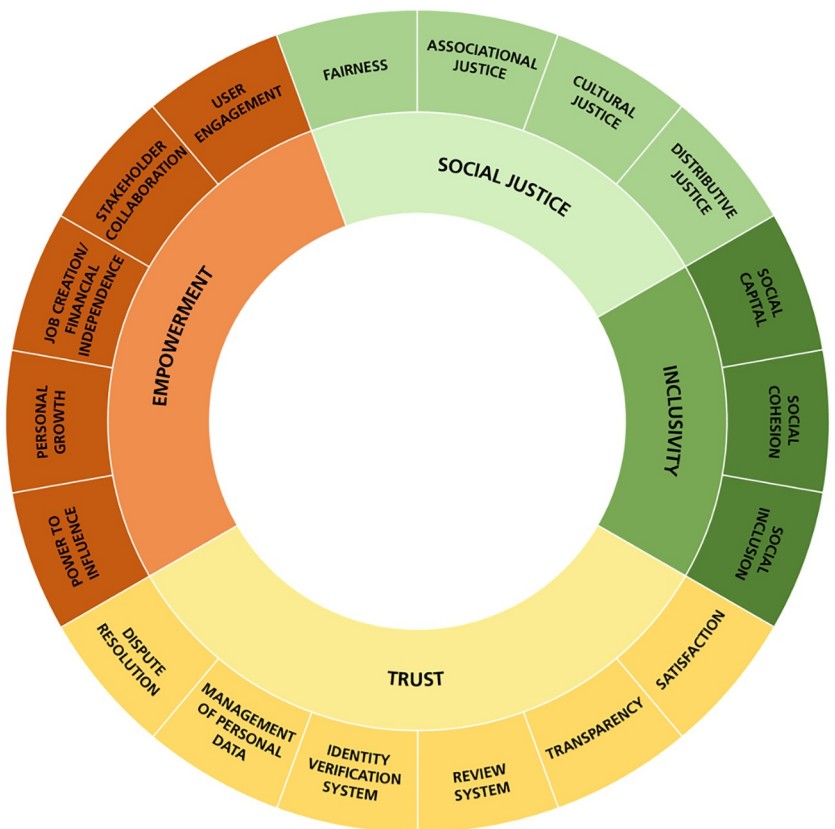

**Fig 2. Systematic framework to assess social sustainability of sharing platforms.**

then provide a practice-oriented tool in which we propose measurable variables and corresponding sources of data to assess each indicator.

## Trust

We define trust as the belief in something or someone based on its characteristics (e.g. personability, ability, performance, integrity, transparency, achievements, or history). Trust, a scarce resource, is identified as a key driver of the sharing economy [97, 98] and a key enabler of transactions in ICT-mediated business models [39, 40]. Trust is particularly important in two-sided markets [20, 99] in the sharing economy, where the platform facilitates an exchange between a resource owner and resource user. The sharing economy has been shown to increase trust between peers [46, 49]. Research distinguishes three types of trust: peer, platform, and product [20].

In their work on trust in the sharing economy, Hawlitschek et al. [20] explore trust in peers among a supplying-peer and a consuming-peer in peer-to-peer markets. We expand the definition of the sharing economy to include any two-sided market, e.g. peer-to-peer, business-to-business, business-to-peer, and crowd/cooperative [6, 28]. Here, we describe trust among platform users—a resource owner and a resource user. This trust among users is maintained as long as each has the *ability* to execute their key activity in co-creating value on the platform with high *integrity* and *benevolence* [20, 100]. These three dimensions—ability, integrity, and benevolence—are well established for gauging trust online [41, 101]. Ability describes the skills and competencies of users; integrity is perceived among users by honouring and upholding their responsibilities and commitments; benevolence considers the actions of users with each other's needs in mind. These dimensions are particularly important in facilitating trust among users, as together they share risks traditionally held by the business in business-to-consumer transactions [39], including economic loss, damage, theft, legal restrictions, and personal safety.

Trust in a platform is also established by these three dimensions, which can increase the likelihood of users continuing patronage on the platform [20, 102]. And, platforms can manifest ability, integrity, and benevolence, depending on their business model choices, e.g. pricing mechanism, review system, revenue streams, see Curtis & Mont [6]. For example, trust in Airbnb as a platform is fostered only when the platform has transparent booking and payment processes, functioning identity verification systems, and adequate data and privacy standards. Thus, the platform itself takes the pivotal role in establishing and maintaining trust among users [45]. Finally, trust in a product describes the belief that the product or service will satisfy the need of the user [20, 103]. As the product is an inanimate object, trust is only fostered through its ability to fulfil its function in terms of quality, durability, and ease of operation.

Based on the three types of trust and the corresponding dimensions of ability, integrity, and transparency, we identify several indicators to measure trust of sharing platforms and their practices: satisfaction, transparency, meaningfulness of a review system and an identity verification system, management of personal data, and dispute resolution.

**Satisfaction.**   Satisfaction describes how users perceive matching or fulfilling of offers and demands [104], reflecting the *ability* of users and products to do so. In the sharing economy, this describes the needs of both the resource owner and resource user being fulfilled with minimum effort. In considering peer-to-peer accommodation sharing, Tussyadiah [105] has identified four significant determinants of satisfaction—enjoyment, economic benefits, amenities, and sustainability—but acknowledges that determinants of satisfaction are likely to differ between service offerings and users. Interestingly, Tussyadiah [105] found that improved sustainability outcomes actually detracted from the user's perception of satisfaction, suggesting

that the majority of users are not motivated by sustainability when engaged in peer-to-peer accommodation sharing. Möhlmann [106] has also shown that cost savings, trust, familiarity, service quality, and utility most significantly influence user satisfaction in the sharing economy. Users also report satisfaction in new social networks developed via platforms like TaskRabbit and Couchsurfing [107–109]. Satisfaction is therefore perceived by the user, relating to their needs and experiences associated with convenience, cost savings, utility, environmental impact, and/or social interactions, among others.

**Transparency.** Navigating the sharing economy, users and municipal actors are hampered by a lack of transparency among sharing platforms like Airbnb [110, 111]. This challenge is exacerbated by the use of digital platforms to facilitate sharing, which may lead to depersonalisation, anonymity, and less transparency among users [23]. Transparency measures are closely tied to trust-generating mechanisms like review systems and identity verification systems [112]. Transparency describes the level of openness in how sharing platforms govern, interact with users, and communicate about how they store or process personal sensitive data. Without platform and user transparency, user safety is jeopardised [112] and the social or environmental claims made by sharing platforms are meaningless [110]. Therefore, transparency is something that sharing platforms must practice (e.g. open reporting, data sharing, communication campaigns) and facilitate among users (e.g. review and identity verification systems).

**Review system.** A review or rating system supports transparency and facilitates trust in users, platforms, and products [36, 112]. Review systems are said to reduce the perceived risk of receiving inferior service quality or social interaction [113]. These systems seek to incentivise both resource owners and resource users to "create a respectful and accommodating demeanour during exchanges" [106, 114] and "decrease the interpersonal trust necessary between [users]" [115]. Review systems also reduce transaction costs normally associated with seeking recommendations and assessing service quality [116].

While they may facilitate trust, user reviews are largely positive and do little to distinguish between service quality or social interaction [117, 118]. Positive reviews may reflect perceived social or cultural capital by some excluding others, for example, based on race, religion, or economic status, which may undermine feelings of empowerment and social justice among users excluded from such interactions [113, 119]. Review systems may also aid discrimination and racial tension [22]. It is necessary to assess the perceived effectiveness of the review system among users as well as known or perceived issues regarding its abuse.

**Identity verification system.** Platforms rely on identity verification to facilitate trust and ensure safety in the social interactions and exchanges taking place on their platforms [120]. Verification varies across platforms, for example, using pictures, uploading official identification documents (e.g. driver's license, passport), verifying email or telephone numbers, or relying on existing users to verify the identity of new users. While intended to foster trust and safety, information about individuals revealed through the verification process—gender, sexual, ethnic, or racial identity—may also lead to issues with safety and discrimination [23]. Furthermore, transparency and management of personal data are essential to ensure this practice is meaningful for facilitating trust [121].

**Management of personal data.** One of the latest additions to the literature on social impacts of the sharing economy is the impacts associated with protection of privacy [72]. A paradox emerges when personal data is necessary for identity verification to facilitate trust [120] while, at the same time, commercial sharing platforms extract, store, and monetise personal data as a source of revenue [10, 30]. Therefore, platforms serve as data controllers and data processors, and it is solely the responsibility of platforms to manage personal data and avoid "data spills" from users sharing personal data in reviews or among each other [122].

Effectively managing the protection of personal data is also a matter of legal responsibility [122]. Assessing management of personal data requires examining the policies and practices of platforms as well as understanding the perception among users regarding its efficacy and transparency.

**Dispute resolution.**   The presence, effectiveness, and fairness of dispute resolution mechanisms may facilitate trust among users. These mechanisms consider problems with compliance, complaints, and user satisfaction [23, 116]. However, sharing platforms commonly rely on "within-platform" resolution mechanisms, which are often not transparent and lead to uneven treatment [23]. We expect to see robust and transparent dispute resolution mechanisms to prevent such behaviour.

## Empowerment

Empowerment generally describes the users' perceived power to influence the service offering and/or decision-making and governance of the platform [123]. In this way, governance refers to the approach by the platform to involve users in decision-making as well as the exchange of benefits among them [6, 124], which influences feelings of empowerment. It is also described as the "…enhanced ability to access, understand and share information" [125]. Sharing platforms are said to have an ideological orientation towards empowering their users [8]. Platforms facilitate empowerment by decentralising modes of consumption [10], generating additional income [114], and providing greater access to goods and services otherwise unattainable via ownership. Technology enhances this ability by providing better networking, communication, and opportunities for collaboration [126]. This is an example of digital empowerment [126], which may also increase self-efficacy and the opportunity to learn new skills [123]. Therefore, empowerment is an important aspect when evaluating sharing platforms [127].

In the sharing economy, resource owners may earn additional or primary income, while resource users are empowered by accessing goods and services they otherwise could not afford via ownership. Empowerment has an enabling aspect, offering control to users traditionally ceded to businesses and suppliers [125].

Empowerment is closely associated with other social aspects such as trust, where users that feel a sense of empowerment are likely to have more trust in the platform [123, 128]. Empowerment is also seen as both a process and an outcome; the former may be influenced by the practices of the sharing platform and the latter based on the subjective experience of the users [125]. We propose several indicators to assess empowerment: power to influence, personal growth, job creation or financial independence, stakeholder collaboration, and user engagement.

**Power to influence.**   Power to influence describes the users' perceived ability to affect the operations of the platform as well as the exchanges and interactions taking place. A more cooperative or collaborative governance model may empower users to exert influence in the day-to-day decision-making of sharing platforms [6]. Reviews or ratings are another mechanism by which users may influence exchanges and interactions on the platform [12]. While specific business model choices may empower users to influence operations and exchanges, any assessment requires measuring the users' perceived power.

Similarly, we extend the power to influence more broadly to society, where stakeholders outside the platform ecosystem (e.g. neighbourhoods, community groups, city councils) may also exert power to influence operations and exchanges. Increasing the social sustainability of sharing platforms must also provide space for societal groups to respond to challenges arising from their activities (e.g. gentrification, housing affordability, discrimination, casualisation of

labour, taxes). The openness of sharing platforms to listen and respond to the concerns or wishes of societal actors is also an important indication of empowerment.

**Personal growth.**   Personal growth reflects opportunity to learn new skills through training, experience, and social interactions [129]. Interaction among users via sharing platforms can develop new social and cultural skills [113]. However, sharing platforms can be more intentional, providing training in the use of the technology, e.g. smartphones, needed before users can take advantage of the platform and its offerings [130], which increases social justice and inclusivity. Workshops and experience sharing among users may also provide the opportunity to learn how to use specific products [131], for example, tools or professional photography equipment.

**Job creation / financial independence.**   The sharing economy is said to foster economic empowerment [55, 132] through job creation, greater income, and increased financial independence [49, 133]. Resource owners can earn money by providing access to goods and services [114, 134]. Studies are emerging that demonstrate how sharing platforms provide real-time flexibility to earnings and potentially lead to higher hourly wages [135]. The sharing economy probably creates more opportunities for employment than it eliminates [116], with users valuing the flexibility in hours and effort they may choose to engage in the platform [14]. Users can earn money by providing access to shared resources or save money by accessing shared resources more cheaply than buying new, leading to financial independence and a sense of empowerment.

However, the perceived job creation and financial independence does not come without challenges. Because of rebound effects and our complex systems of production and consumption, it is difficult to determine net jobs created or the impact of secondary consumption as a result of savings in the sharing economy [25, 36, 51, 110]. In times of economic crisis, users that rely on revenues generated from the sharing economy lack legal protections and employment contracts compared to traditional employment [14]. Many authors warn that the lack of regulation and labour unions can also lead to precarious employment and poor working conditions [73, 74]. Any assessment of perceived economic empowerment in the good times must also be balanced with the potentially devastating personal and societal economic consequences in the bad times.

**Stakeholder collaboration.**   Stakeholder collaboration is closely tied to other modes of empowerment such as power to influence, and other social aspects like trust and inclusivity. We describe stakeholder collaboration as the willingness or openness of sharing platforms to involve others in the design and implementation of their offering, which can be an important motivating factor for resource owners, resource users, and societal actors to feel a sense of empowerment. Collaboration is also an important mechanism to build and maintain reputation [136]. This is a platform-level indicator, but assessing this indicator can be triangulated with stakeholders' perception of their ability to collaborate with sharing platforms.

**User engagement.**   We propose that high levels of user activity and engagement demonstrate a sense of empowerment in those using the platform. User activity and engagement may be measured by, for example, the number of transactions, the length of use or membership, and involvement in governance. This indicator may be related to satisfaction, since users whose needs are fulfilled are likely to continue to engage with the platform.

## Social justice

Some authors argue that the sharing economy contributes to social justice [55, 98, 137], but more research is needed to operationalise the concept in the context of the sharing economy and to describe the specific mechanisms that may enhance social justice. The term relates to

issues of equity, defined by Young [138] as ". . .the morally proper distribution of social benefits and burdens among society's members". However, this "distributive paradigm" over-emphasises access, ignoring the existing institutional and social structures that lead to inequitable distribution in the first place [138, 139]. Building on the work by Fraser [140] and Young [138], Cribb and Gerwen [141] propose the dimensions of social justice—distributive, cultural, and associational justice—which we complement with the additional indicator of fairness.

**Distributive justice.**   Distributive social justice includes material goods—wealth, income, resources—and nonmaterial goods—rights, opportunity, power, and dignity [138, 139]. This indicator is closely associated with other indicators, including personal growth and social inclusion. To strive for distributive justice, efforts to minimise or completely eliminate exploitation, marginalisation, or deprivation are necessary [140, 141]. Sharing platforms may be assessed based on their practices to remedy existing inequitable distribution as well as to mitigate reinforcing inequitable structures. For example, access to technology ensures users have the opportunity to access resources on the platform. Business model choices involving governance (e.g. cooperative) and value orientation (e.g. economic, environmental, social, societal) also open up for more equitable sharing of economic and noneconomic benefits [6]. However, evidence and experience show that this is an area in which the sharing economy can improve. Findings presented by Piracha [23] show that "sharing platforms align with neoliberal impulses, to roll-back laws and regulations that provide safeguards for sections of society from economic exploitation and discrimination".

**Cultural justice.**   Cultural justice promotes the recognition, representation, and tolerance of different cultures and communities, not limited to ethnic or racial cultures [142]. Fraser [140] says cultural justice must preclude domination, non-recognition, or disrespect by any other social or cultural group, often in the majority. Cultural justice is achieved in parallel with other indicators promoting personal growth and social inclusion. While recognition supports users from vulnerable or marginalised groups, their representation can also reward organisations that include these groups in governance, by learning new practices, accessing new markets, and enhancing the diversity of social interaction among users [141].

In the pursuit of cultural justice, it is important to include representation from groups when decisions are made in relation to those groups [142]. Cultural awareness is important when operating in new communities. For example, Boateng et al. [114] states that ". . .the sharing economy, in general, can impact negatively on collective and hospitable societies such as Ghana. That is, although Uber and the sharing economy, in general, have some social benefits, they also have some negative social-cultural effects". The criticism made by Boateng et al. [114] and Haerewa et al. [143], among others, is that the practice of sharing platforms must not undermine the cultural practices of the communities in which they operate.

**Associational justice.**   Associational justice—also referred to as participatory justice—seeks to include marginalised groups in the decision-making processes that impact their experiences [142]. Associational justice is a prerequisite to achieving distributive and cultural justice, as this requires representation and participation [141]. However, assessing associational justice is difficult; the presence of participation pathways is not sufficient to overcome the distributive and cultural injustices entrenched in society [141]. This indicator is closely associated with power to influence, stakeholder collaboration, and social inclusion, but describes the equity and fairness in participation on the platform.

**Fairness.**   Common to these three indicators is the perceived fairness of material and non-material distribution (distributive justice), representation (cultural justice), and participation (associational justice) among users of the sharing platform. Fairness is a somewhat vague term and is perceived by the group which is acting or being acted upon. However, it is described as a social value of sharing platforms that must be achieved to ensure a socially sustainable

sharing system [144]. The perception of fairness is also important, to influence public accept-ability of sharing platforms and their activities [145].

Many of the platforms in the sharing economy promote some degree of access, democrati-sation, openness, inclusivity, and/or equality. However, Schor et al. [8] find significant evi-dence of "distinguishing practices" based on class and power, which subvert the values of fairness prescribed by platforms. Therefore, assessing fairness must be balanced between the stated values of the platform and the perceived fairness among users. One area where this is most relevant is dispute resolution, when users may perceive the experience as more or less fair if the mechanism is transparent or just.

## Inclusivity

In literature, inclusivity is a vague concept that captures many different social activities such as inclusion, connectedness, and the quality of interaction. To place this term in context, the sharing economy is said to foster "inclusive growth" [36, 130, 146], a nebulous term to describe both the outcome and process that seeks to enfranchise individuals and communities during economic opportunities [147]. As a process, mechanisms that enable participation, e.g. gover-nance, ownership, employment, consumption, risk/reward, are important when considering inclusivity [147]. Inclusion is also interconnected with the other indicators and aspects: trans-parency, stakeholder collaboration, and associational justice.

Oxoby [148] provides the most convincing description of inclusion and its related concepts, defining inclusion as a process that provides "equal access to rights and resources" as well as the elimination of barriers to participation [148]. We propose three broad indicators to assess inclusivity: social inclusion, social cohesion, and social capital. Again, Oxoby [148] describes their interconnectedness: social capital describes an individual's resources (e.g. time, effort, assets) invested during interaction; social cohesion is the accumulated social capital, a charac-teristic of the group/economy/society; and social inclusion is the mechanism that increases the opportunity and desire to invest social capital. In other words, social capital is the flow, social cohesion is the stock, and social inclusion is the process as well as the outcome. If all these three elements come together, we can speak of an inclusive sharing economy that can integrate all its diverse members.

**Social capital.** According to Portes [149], the first systematic definition of social capital was provided by Bourdieu [150], who describes social capital as ". . .the aggregate of the actual or potential resources which are linked to possession of a durable network of more or less institutionalized relationships of mutual acquaintance or recognition". More recently, social capital was expanded by Berger-Schmitt [151] to include the interaction and engagement within social group(s), the quality of social interaction, and the quality of the supported/sup-porting societal institutions. Components such as willingness to participate, willingness to cooperate, and sense of belonging are also characteristics of social capital [152–154]. However, the definition of social capital is said to have been "independently invented at least six times" during the 20th century [155]. Therefore, similar to Bourdieu [150] and later Oxoby [148], we define social capital as the stock of an individual's resources (e.g. time, effort, assets) invested during interaction, where the accumulation informs the quality of interaction and related institutions. Investment of social capital requires adequate trust; therefore, trust is also an important factor in building social capital [148].

By meeting new people, engaging with others, and increasing social interactions, the shar-ing economy is said to build social capital [51, 106, 156], so interactions between people are needed [157]. However, beyond stating that the sharing economy may foster social capital, this is under-examined in literature [120]. Nonetheless, the accumulation of social capital is said to

provide benefits, including civic engagement, economic prosperity, and improved outcomes for individuals—e.g. health, happiness, well-being—and society—e.g. institutions, safety, community [155, 157].

**Social cohesion.** Social cohesion is a characteristic of a group, economy, or society, generated by accumulated social capital [148, 158]. The concept describes the cumulative effect of establishing social ties among people who take part in the practice of sharing. Several studies identify a strong positive contribution to social capital and social cohesion [105, 108, 115, 120, 159–161]. However, in the case of Airbnb, these ties are only built if the host and guest interact, e.g. if single rooms are rented out instead of the whole apartment [105]. In addition, the level of technical involvement has an influence on social interaction, as the effect decreases as technology becomes more developed [115]. Studies find a relationship between the monetisation of sharing practices and the development of social ties—the likelihood of building new ties is greater when the consumption practice is non-profit and local [145, 159]. Similarly, social belonging has been described in studies on ride-sharing, land-sharing, and peer-to-peer insurance platforms [143, 162–164]. Closely associated with cultural justice, social cohesion can also describe the accumulation of cultural learning and cosmopolitan capital [120, 161].

However, these positive impacts are not always observed. Several studies also highlight the missing or negative impact of sharing on social cohesion. Users of Airbnb and Uber often have little to no interest in social interaction [105, 114, 159, 165]. Accommodation sharing can also reduce the sense of community within cities [53, 56].

**Social inclusion.** Social inclusion describes "…the extent that individuals, families, and communities are able to fully participate in society and control their own destinies, taking into account a variety of factors related to economic resources, employment, health, education, housing, recreation, culture, and civic engagement" [166]. While literature suggests the sharing economy may foster social inclusion as an outcome [7, 98, 167], the processes by which this takes place are underexplored.

Research has highlighted the experiences of social *exclusion* among users in the form of discrimination or bias based on race, ethnicity, gender identity, sexual orientation, religion, class, or digital literacy, among others [15, 22, 23, 27, 120]. Studies reveal cases of racial discrimination [22, 71], digital discrimination [69], and ethnic discrimination [70]. For example, one study explored the correlation between socio-demographic parameters of tenants and geographical location of Airbnb listings [111], while another connected the location of free-floating carsharing vehicles to certain demographic groups [168]. Research indicates that advantaged populations, i.e. white, young, well-educated, and employed, disproportionately reap the benefits facilitated by sharing platforms [168]. Therefore, while social inclusion is promoted as an outcome of the sharing economy, empirical evidence and individual experience say otherwise.

Sharing platforms have responded by asserting that users may not decline service to any other user on the basis of protected class; however, this has been criticised as outsourcing responsibility to users—instead of the sharing platform—to ensure social inclusion [120]. In any assessment, we suggest the need to identify the specific mechanisms or practices used by sharing platforms to promote social inclusion. These likely vary according to business model and cultural or geographical context. However, Ladegaard [120] suggests making it more difficult to determine the race, location, or socioeconomic status of users, suggesting that pictures are not necessary if substituted with a meaningful review system. Platforms have implemented anti-discrimination training programmes for users and dispute resolution mechanisms to respond to complaints of discrimination [23]. Any effort to foster a sense of inclusivity must be balanced with mechanisms to foster trust, empowerment, and user safety on the platform; more intimate exchanges may require greater information available to users to ensure safety.

## Operationalising the framework

Through the process of testing and validating the framework, we identified the need to operationalise the framework in the form of a tool. The framework provides a structure to classify, categorise, and assess the social impacts of sharing platforms. However, to increase relevance for sharing platforms and other interested actors, we propose a practice-oriented tool that provides measurable variables for each of the four aspects across the eighteen indicators (Table 5). We developed the tool based on the above framework, synthesising inputs from literature and the stakeholder workshop as well as our own contributions. The measurable variables should be seen as suggested data points to help inform the social sustainability assessment of sharing platforms.

Prior to using the tool, we suggest defining the purpose for assessing the social sustainability of a sharing platform. Aspect(s) or indicator(s) could be chosen for prioritisation on the basis of the purpose or stated values of those using the tool. Then, the sources of data are varied, often affording the opportunity for triangulation. When using the tool, the level of ambition in data collection should be matched with the purpose for using the tool. Furthermore, we suggest triangulating data in relation to other social aspects because of overlapping concepts and cause-effect relationships (Table 6).

Finally, the tool is practice-oriented, intended for use by researchers and practitioners assessing the social impact of sharing platforms. We suggest the tool may be useful to structure data, to illuminate hotspots for sharing platforms to focus their activities, to inform regulation in safeguarding users and society, and to advise investment decisions. However, depending on the purpose, we suggest that the greater the amount of effort, data, variables, and triangulation, the more representative the assessment of the social performance of a sharing platform.

We also recommend caution be applied regarding the process of collecting, storing, or communicating data on the social impact of a sharing platform: 1) collecting data may risk exclusion; and 2) storing and communicating data may lead to data protection issues. Therefore, those using the tool must ensure inclusion of all actors impacted by the activities of the platform (a form of cultural justice in itself) and protect data from improper use.

## Discussion and conclusions

Our society is facing numerous social challenges stemming from increased inequality [169], a growing sense of social distance as a result of technology [170], and yet-unknown impacts from COVID-19. We must respond as individuals, organisations, institutions, and society. In view of the growing concern about adverse effects of sharing platforms, there is a need to mitigate negative social impacts caused by sharing platforms and the practices of their users.

Combining literature and stakeholder perspectives, we develop a systematic framework and practice-oriented tool assessing social impacts of sharing platforms. The proposed framework provides an overview of potential social impacts of sharing platforms and their users. It comprises four main aspects—trust, empowerment, social justice, and inclusivity—and eighteen indicators described in detail in relation to the sharing economy. The framework is then expanded into a practice-oriented tool for researchers and practitioners as a method to assess social impact of sharing platforms.

### Key insights and contributions

Literature and empirical insights suggest numerous adverse social impacts resulting from practices among sharing platforms and their users. We highlight both the potential positive and negative social impacts, recognising that any judgement requires an evidence-based assessment. The aim of our research was: 1) to improve understanding of the social impacts of

**Table 5. Practice-oriented social impact tool with measurable variables.**

| Aspect / Indicator | Measuring Variables | | | |
|---|---|---|---|---|
| **Trust** | **Platform** | **Resource Owner** | **Resource User** | **Society** |
| Satisfaction | • Mechanism to measure user satisfaction (IwSP)<br>• Number of users or frequency of use (as a proxy) (SPD)<br>• Number of resources that are lost, disappeared, or broken (SPD) | • Degree of satisfaction with the service provided by the platform (US)<br>• Degree of satisfaction of the returned resource (US) | • Degree of satisfaction with the service provided by the platform (US)<br>• Degree of satisfaction of the provided resource (US) | N/A |
| Transparency | • Certification by an accredited body (e.g. GRI, B Corp) (IwSP)<br>• Open data practices, several examples (IwSP)<br>• Communication about open data practices reaching at least 50% of users (SPD)<br>• Transparency of platform communication (contact us, dispute resolution)<br>• Transparency in access to information (governance)<br>• Transparency about environmental, social, and economic impacts<br>• Sharing data with local, regional, national governments | • Knowledge of open data practices by platform (Yes/No) (US)<br>• Perceived level of transparency by sharing platform (US)<br>• Perceived level of transparency by resource user (US) | • Knowledge of open data practices by platform (Yes/No) (US)<br>• Perceived level of transparency by sharing platform (US)<br>• Perceived level of transparency by resource owner (US) | • Knowledge of open data practices by platform (Yes/No) (CS, Iw3)<br>• Access to data (and environmental, social, and economic impacts) (Iw3) |
| Review System | • Presence of a review system (IwSP)<br>• Evaluation of perceived effectiveness of review system by third party (IwSP)<br>• Number of reviews compared to number of transactions (SPD) | • Perceived meaningfulness of review system to illustrate:<br>1. platform service<br>2. interaction (IwRO, US) | • Perceived meaningfulness of review system to illustrate:<br>1. platform service<br>2. interaction<br>3. resource quality (IwRU, US) | • Perceived meaningfulness of review system for:<br>1. using<br>2. supporting<br>3. investing<br>4. regulating (CS, Iw3) |
| Identity Verification System | • Presence of an identity verification system<br>1. pictures<br>2. ID documents<br>3. email<br>4. phone number<br>5. existing user verification (SPD, IwSP) | • Perceived meaningfulness of identity verification system (IwRO, US) | • Perceived meaningfulness of identity verification system (IwRU, US) | • Perceived meaningfulness of identity verification system to protect public safety (Iw3, CS) |
| Management of Personal Data | • Mechanisms to protect personal data (IwSP)<br>• Communication about how personal data is collected, processed, stored, and used by the platform or other parties (SPD, IwSP) | • Perceived trust in platform to manage the following in RO's best interest: Personal Data; Personal Identity; Financial Data; Physical Safety (IwRO, US) | • Perceived trust in platform to manage the following in RU's best interest: Personal Data; Personal Identity; Financial Data; Physical Safety (IwRU, US) | • Perceived trust in platform to manage the following in society's best interest: Personal Data; Personal Identity; Financial Data; Physical Safety (CS) |
| Dispute Resolution | • Presence of codes of conduct, or similar (SPD)<br>• Presence of mechanisms to facilitate efficient dispute resolution SPD)<br>• Perceived extent rules are followed by users (IwSP)<br>• Number of disputes filed (SPD) | • Perceived fairness of dispute resolution, if applicable (US) | • Perceived fairness of dispute resolution, if applicable (US) | N/A |
| **Empowerment** | | | | |
| Power to Influence | • Governance model (SPD, IwSP)<br>• Willingness to respond to the concerns of users and societal actors (IwSP) | • Perceived access/control/power to influence the operations of the platform (IwRO, US)<br>• Perceived sense of contribution, ownership (IwRO, US) | • Perceived access/control/power to influence the operations of the platform (IwRU, US)<br>• Perceived sense of contribution, ownership (IwRU, US) | • Perceived openness of platform to respond to the concerns or wishes of societal actors (Iw3, CS) |

*(Continued)*

**Table 5.** (Continued)

| Aspect / Indicator | Measuring Variables | | | |
|---|---|---|---|---|
| **Trust** | **Platform** | **Resource Owner** | **Resource User** | **Society** |
| Personal Growth | • Presence of initiatives to foster personal growth (e.g. trainings, workshops, experience sharing) (SPD, IwSP) | • Perceived opportunity to learn new skills through training, experience, and social interactions (IwRO, US)<br>• Acquired knowledge and skills (e.g. managing finances, social media, communication, marketing, photography, using tools and software) (US) | • Perceived opportunity to learn new skills through training, experience, and social interactions (IwRU, US)<br>• Acquired knowledge and skills (e.g. managing finances, social media, communication, marketing, photography, using tools and software) (US) | N/A |
| Job Creation / Financial Independence | • Financial flows, jobs created (SPD) | • Amount of additional earnings (US)<br>• Perceived change in financial independence, if any (US) | • Amount of money saved (US)<br>• Perceived change in financial independence, if any (US)<br>• Perceived access to new resources (IwRU, US) | • Impact on incumbent industries (Iw3, PD)<br>• Net jobs created/lost in society (PD) |
| Stakeholder Collaboration | • Willingness to involve others in the design and implementation of the platform (IwSP) | • Perceived openness of the platform to collaborate (IwRO, US) | • Perceived openness of the platform to collaborate (IwRU, US) | • Perceived openness of the platform to collaborate (Iw3, CS) |
| User Engagement | • Extent of participation or engagement in governance of the platform (SPD, IwSP)<br>• Number of people using the service (as a proxy) (SPD)<br>• Length of use / membership (SPD)<br>• Presence of initiatives fostering empowerment (e.g. forums, trainings, events) (SPD) | • Perceived meaningfulness of platform initiatives fostering empowerment (e.g. forums, trainings, events) (IwRO, US) | • Perceived meaningfulness of platform initiatives fostering empowerment (e.g. forums, trainings, events) (IwRU, US) | N/A |
| **Social Justice** | | | | |
| Distributive Justice | • Mechanisms for distribution of economic and noneconomic benefits among users, society (SPD, IwSP)<br>• Effort to reduce or eliminate exploitation, marginalisation, or deprivation (e.g. dispute resolution) (IwSP)<br>• Actions to remedy or mitigate inequitable distribution of material and nonmaterial goods (e.g. equal access to goods and services) (IwSP) | • Perceived effectiveness of mechanisms by the platform to enable a more equitable distribution of economic and noneconomic benefits (US)<br>• Perceived effectiveness of actions by the platform to enable a more equitable distribution of material and nonmaterial goods (US) | • Perceived effectiveness of mechanisms by the platform to enable a more equitable distribution of economic and noneconomic benefits (US)<br>• Perceived effectiveness of actions by the platform to enable a more equitable distribution of material and nonmaterial goods (US) | • Reduced reliance on social support (PD)<br>• Perceived effectiveness of mechanisms by the platform to enable a more equitable distribution of economic and noneconomic benefits (US)<br>• Perceived effort to reduce or eliminate exploitation, marginalisation, or deprivation (CS) |
| Cultural Justice | • Representation of different socio-economic groups and under-represented groups in decision-making (IwSP)<br>• Measure the tolerance of different cultures & communities among users (IwSP, SPD)<br>• Mechanisms to reduce bias and discrimination among platform, its users (IwSP)<br>• Ensure cultural practices of the community where sharing occurs are not undermined (IwSP) | • Perceived tolerance, bias, or discrimination during the course of sharing (IwRO, US)<br>• Perceived effectiveness of the platform to protect cultural practices of the community where sharing occurs (US) | • Perceived tolerance, bias, or discrimination during the course of sharing (IwRU, US)<br>• Perceived effectiveness of the platform to protect cultural practices of the community where sharing occurs (US) | • Perceived representation of different socio-economic groups and under-represented groups (CS) |
| Associational Justice | • Participation pathways that ensure representation and distribution of resources (IwSP)<br>• Accessibility (e.g. language, contact us, flexible opening hours) | • Perceived effectiveness of participation pathways, especially overcoming structural and cultural injustices (IwRO, US) | • Perceived effectiveness of participation pathways, especially overcoming structural and cultural injustices (IwRU, US) | • Perceived effectiveness of participation pathways, especially overcoming structural and cultural injustices (Iw3, CS, M) |
| Fairness | • Perceived fairness of platform activities in distribution, representation, and participation based on social or cultural class (IwSP) | • Perceived fairness of platform activities in distribution, representation, and participation based on social or cultural class (IwRO, US) | • Perceived fairness of platform activities in distribution, representation, and participation based on social or cultural class (IwRU, US) | • Perceived fairness of platform activities in distribution, representation, and participation based on social or cultural class (Iw3, CS, M) |

(*Continued*)

**Table 5.** (Continued)

| Aspect / Indicator | Measuring Variables | | | |
|---|---|---|---|---|
| **Trust** | **Platform** | **Resource Owner** | **Resource User** | **Society** |
| **Inclusivity** | | | | |
| Social Inclusion | • Measures to promote the opportunity to participate in the activities of the platform (IwSP)<br>• Mechanisms to safeguard review and identity verification system from bias or discrimination among users (IwSP)<br>• Anti-discrimination trainings (SPD)<br>• Dispute resolution mechanisms to deal with issues of exclusion (IwSP)<br>• Number of transactions by e.g. age, race, gender, proportional to society | • Perceived effectiveness of platform measures to promote the opportunity to participate in the activities of the platform (IwRO, US) | • Perceived effectiveness of platform measures to promote the opportunity to participate in the activities of the platform (IwRU, US) | • Perceived effectiveness of platform measures to promote the opportunity to participate in the activities of the platform (Iw3, CS)<br>• Possibility of citizens (i.e. non-users) to make suggestions or participate in dialogue |
| Social Cohesion | • Practices to promote forming of new relationships (SPD, IwSP)<br>• Demonstrated awareness of platform impact on social ties among its users and community (IwSP) | • Perceived degree of interaction during the practice of sharing (US)<br>• Evidence of forming new relationships (IwRO, US)<br>• Perceived strength of social ties within sharing community (IwRO) | • Perceived degree of interaction during the practice of sharing (US)<br>• Evidence of forming new relationships (IwRU, US)<br>• Perceived strength of social ties within sharing community (IwRU) | • Perceived impact of platform activities on the sense of community (Iw3, CS, M) |
| Social Capital | • Prioritises trust-building mechanisms to promote interaction (IwSP) | • Perceived time, effort, resources invested in sharing on the platform (IwRO)<br>• Perceived quality of interactions on the platform (IwRO, US)<br>• Improved personal outcomes (e.g. health, happiness, well-being) (IwRO, US) | • Perceived time, effort, resources invested in sharing on the platform (IwRU)<br>• Perceived quality of interactions on the platform (IwRU, US)<br>• Improved personal outcomes (e.g. health, happiness, well-being) (IwRU, US) | • Perceived impact of platform activities on civic engagement, economic prosperity, consumer safety, and societal institutions (e.g. public transport, media) (Iw3, PD, CS) |

**Proposed data sources**: citizen survey (CS), interview with resource owner (IwRO), interview with resource user (IwRU), interview with sharing platform (IsSP), interview with society actors (e.g. citizens, investors, regulators, and municipal actors) (Iw3), media (e.g. newspapers, blog posts, social media) (M), public data (PD), sharing platform data (SPD), user survey (US).

sharing platforms; 2) to develop a systematic social sustainability framework to structure assessment of sharing platforms; and 3) to operationalise the framework by proposing a tool to support assessment of the social impacts of sharing platforms.

Our work highlights the interrelationship and interconnectedness of platform and user practices, as well as their subsequent social impacts. The social aspects and indicators presented in the framework are closely interrelated through intricate cause-effect relationships. For instance, private earning or savings contribute to empowerment and issues of inclusivity and social justice. Transparency not only builds trust among stakeholders, but also frames conditions for increased inclusivity and social justice. We explicate these interrelationships in the framework and the practice-oriented tool. While increasing the complexity of assessing social impact, the interconnectedness allows for triangulation of data during assessment as well as the compounding of social benefits if sharing platforms introduce specific mechanisms to overcome adverse impacts.

We contribute to research on understanding and assessing the social impact of sharing platforms in several ways. Firstly, the framework and subsequent practice-oriented tool is holistic and comprehensive in its design and operationalisation. Instead of taking a single perspective, it integrates insights from other studies on trust [20], discrimination [22, 23], social inclusion [146], for example. By providing detailed descriptions for each indicator, the framework is more easily operationalised, facilitating assessment of the diverse social impacts systematically to describe the overall social performance of a sharing platform.

**Table 6. Social indicator relationships for triangulation.**

| Aspect / Indicator | Relation to Other Social Aspects | | | |
|---|---|---|---|---|
| | Trust | Empowerment | Inclusivity | Social Justice |
| **Trust** | | | | |
| Satisfaction | | x | | |
| Transparency | | | x | x |
| Review System | | x | x | x |
| Identity Verification System | | x | | x |
| Management of Personal Data | | x | | x |
| Dispute Resolution | | | x | x |
| **Empowerment** | | | | |
| Power to Influence | x | | | |
| Personal Growth | | | x | |
| Job Creation / Financial Independence | | | | x |
| Stakeholder Collaboration | x | | | |
| User Engagement | x | | | |
| **Inclusivity** | | | | |
| Social Inclusion | | x | | x |
| Social Cohesion | x | | | |
| Social Capital | x | | | x |
| **Social Justice** | | | | |
| Distributive Justice | | x | x | |
| Cultural Justice | | x | x | |
| Associational Justice | | x | x | |
| Fairness | | x | x | |

Additionally, stakeholders participated in defining aspects and indicators of the framework, prioritising aspects based on their perspectives and experiences. The framework not only incorporates these views, we also develop a tool for use by many of these same stakeholders.

Finally, the framework provides increased granularity and decomposition of social impacts relevant to sharing platforms. This is addressed in two ways: the detailed resolution of the framework and the incorporation of actors' views in assessing social impact. Our framework provides detailed descriptions for each of the aspects and indicators, and discusses their relevance to the sharing economy. Our framework and practice-oriented tool also recognise that the relevance of social indicators varies according to the perspective and experiences of the actors involved or impacted by sharing platforms. This unique approach enables flexible use of the framework and tool, depending on the purpose, viewpoints, and priorities of those using the tool. This flexibility also allows for adaptation across sharing platforms, as there are considerable differences based on, for example, the shared practice (e.g. shared mobility, shared goods) or platform type (e.g. peer-to-peer).

## Implications for research and practice

One of the struggles we faced in developing our framework was the vague concepts used in research to describe the various social impacts. While some concepts have more or less established definitions, e.g. gentrification and discrimination, others lack clear boundaries or are used interchangeably. Our framework seeks to provide clearer demarcations of these fuzzy concepts, for example, by describing social capital as the flow, social cohesion as the stock, and social inclusion as the process. Not only does our framework advance research on

understanding and assessing the social impacts of sharing platforms, we hope it also has implications for how others use these concepts. Our framework may primarily be used by researchers to improve understanding of the potential social impacts of sharing platforms and to structure future assessments. However, researchers will find they must collaborate with platforms, which maintain access to their data, prioritise their own impacts, and adapt their business model choices and offerings to enhance social value creation.

Our tool is also intended for use by practitioners—including sharing platforms, governments, investors, and other interested parties—to structure their assessment or understanding of the social impact of sharing platforms. However, while many tools have been created by academia and industry, there is little evidence to suggest these tools are put to use [171]. Research suggests that these tools are often not adapted to meet the specific needs and expectations of companies [172, 173], and tools may remain unused because they are too complex, too demanding of time and resources, or too context specific [172, 174]. Finally, tools that have not involved key stakeholders in their development may miss key insights detracting from their relevance [175].

We responded to these common shortcomings when developing the tool. First, we included stakeholders in the design and description of the aspects and indicators, with their perspectives represented in the tool. We suggest ways in which the tool can be adapted to the needs and purposes of those using it, particularly prioritising aspects and indicators. Finally, we sought to make the tool easier to use than the intricate framework by suggesting measurable variables and sources of data.

## Limitations and future research

Our framework attempts to provide a holistic assessment framework, capturing the breadth of social impacts, experiences, and practices within the sharing economy. While we seek to balance granularity, flexibility, and level of detail, we wish to recognise some limitations of our work in doing so. We recognise that our own perspectives and experiences influence our interpretation of literature and data. By incorporating stakeholder perspectives, we sought to capture greater insights, but the stakeholder workshop involved primarily Swedish participants and captured viewpoints of only those able to attend a single event in person. While the stakeholders included companies, special interests, municipalities, public authorities, and academia, there was no specific representation of platform users or citizens in general. To address this, we encouraged participants at the workshop to consider their perspectives as users as well as citizens. However, we encourage additional testing of our framework and tool in additional national or cultural contexts with relevant stakeholders. Most likely, the range of social impacts and assessment techniques will differ drastically according to socio-cultural, economic, technological, and regulatory contexts. This includes prioritisation of certain social aspects and indicators over others, which is value-laden and requires explicit transparency when using the framework. While the framework and tool are intended to be flexible, based on priorities, purposes, and access to data, it is not yet known how this would impact the comparability of assessment results.

We propose that future research use the framework to compare social impact across these contexts or business models. For example, studies may compare the social impacts between shared practices (e.g. shared space, shared mobility, shared goods), platform type (e.g. peer-to-peer, business-to-consumer), geographical scope (e.g. existing community, local, regional, national, international) and value orientation (e.g. commercial, environmental, social). The extent to which, and how, these business model choices affect the type and scale of social

impacts should be empirically tested. In doing so, particular business model choices may be exemplified as creating, preserving, undermining, or destroying social value.

We suggest in-depth analyses of several sharing platforms to understand the potential interlinkages of impacts and their causalities. This is important, for example, to understand the subsequent impact pathways. In addition, we find that some practices increase social impacts, but diminish others. For example, review and identity verification systems can increase trust and safety, but also lead to discrimination based on race, gender, or disability. While the framework and tool are diagnostic, they are not necessarily prognostic; this could be improved by identifying interlinkages, causalities, and impact pathways.

Our framework seeks to fulfil a stated need by research and practitioners to assess the social impact of sharing platforms. If we do not systematically measure social impacts of sharing platforms, the positive impacts may be overlooked, as a result of increasing focus on the negative impacts, leading to reluctance or cynicism towards sharing in general [130, 132, 176]. In addition, sharing platforms have expressed both the interest and the need to be able to measure their sustainability impacts [24, 177], to communicate with their users, defend their activities among regulators, and secure funding from financiers. We hope this framework and practice-oriented tool may support future research and inspire improved practices to promote a more positive social impact of sharing platforms.

## Supporting information

**S1 Appendix. Keywords identified during preliminary literature review.**
(DOCX)

**S2 Appendix. Social aspects and impacts from preliminary literature review.**
(DOCX)

**S3 Appendix. Definitions of social aspects presented at workshop.**
(DOCX)

**S4 Appendix. Workshop data.**
(DOCX)

## Acknowledgments

We would like to thank Charlotte Liere and Kes McCormick, project leaders of Sharing Cities Sweden, which provided the opportunity to organise the stakeholder workshop. We would also like to thank participants of the stakeholder workshop and those that provided feedback on the framework at the 6th International Workshop on the Sharing Economy and the Nordic Sharing Cities Summit. We would like to thank Yuliya Voytenko Palgan for valuable support and feedback during the writing process.

## Author Contributions

**Conceptualization:** Steven Kane Curtis, Jagdeep Singh, Oksana Mont.

**Data curation:** Steven Kane Curtis, Jagdeep Singh, Oksana Mont, Alexandra Kessler.

**Formal analysis:** Steven Kane Curtis, Jagdeep Singh, Oksana Mont.

**Funding acquisition:** Oksana Mont.

**Methodology:** Steven Kane Curtis, Jagdeep Singh, Alexandra Kessler.

**Project administration:** Steven Kane Curtis, Oksana Mont.

**Supervision:** Oksana Mont.

**Validation:** Steven Kane Curtis, Jagdeep Singh.

**Visualization:** Steven Kane Curtis, Jagdeep Singh.

**Writing – original draft:** Steven Kane Curtis, Oksana Mont, Alexandra Kessler.

**Writing – review & editing:** Steven Kane Curtis, Jagdeep Singh, Oksana Mont, Alexandra Kessler.

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
