## [Decision Letter · Decision Letter 0]

10 Jul 2020

PONE-D-20-15696

Systematic framework to assess social impacts of sharing platforms: Synthesising literature and stakeholder perspectives to arrive at a framework and practice-oriented tool

PLOS ONE

Dear Authors,

Thank you for submitting your manuscript to PLOS ONE. After careful consideration, we feel that it has merit but does not fully meet PLOS ONE’s publication criteria as it currently stands. Therefore, we invite you to submit a revised version of the manuscript that addresses the points raised during the review process.

Please see comments below.

We look forward to receiving your revised manuscript.

Kind regards,

Dejan Dragan, PhD

Academic Editor

PLOS ONE

Additional Editor Comments:

Major revision is required. The AE

Journal Requirements:

2. Please amend either the title on the online submission form (via Edit Submission) or the title in the manuscript so that they are identical.

Reviewers' comments:

Reviewer's Responses to Questions

**Comments to the Author**

1. Is the manuscript technically sound, and do the data support the conclusions?

Reviewer #1: Partly

Reviewer #2: Partly

Reviewer #3: Yes

2. Has the statistical analysis been performed appropriately and rigorously? 

Reviewer #1: N/A

Reviewer #2: No

Reviewer #3: Yes

3. Have the authors made all data underlying the findings in their manuscript fully available?

Reviewer #1: Yes

Reviewer #2: No

Reviewer #3: Yes

4. Is the manuscript presented in an intelligible fashion and written in standard English?

Reviewer #1: Yes

Reviewer #2: Yes

Reviewer #3: Yes

5. Review Comments to the Author

Reviewer #1: PLOS ONE

Systematic framework to assess social impacts of sharing platforms: Synthesising literature and stakeholder perspectives to arrive at a framework and practice-oriented tool.

The paper aims to make an analysis of the “sharing platforms” and their social impact.

The idea is attractive and novel, the document is worked and is widely documented. The sources of information have value in itself, and the goal of setting a set of measurement variables is interesting.

The work deals with a lot of aspects that are very interrelated and that are generally little concretes. This makes the work very long. From my point of view, the paper must be written in a clearer and more concrete way making an effort to show the reader the relationship between the concepts managed in a profound and concrete way. The Workshop and the table 5 seem to be especially valuable.

New communication technologies (ICTs) allow for a reduction in costs that are the basis of the “sharing economy”. Technological advances in communication generate value in a transversal way and one way in which it produces value is by reducing transaction costs which increases the supply and allows access to the service at a lower price and generates social wealth since it is consumers who appropriate the value generated. The cost reduction is a consequence of: 1) increasing the level of information before and after the contract (adverse selection and moral hazard) (what at work has been called “trust”). Increased information also reduces “residual loss”, giving access to demand cheaply to the producer and giving access to more supply by the consumer; 2) reduces limited rationality, facilitating information management by grouping it into satisfaction indicators that allow for easy interpretation (likes, stars,...).

When transaction costs are high, the company replaces the market as a form of concentration of resources in order to save the high costs of marketing the product. The aforementioned transaction cost reduction allows the market (individual persons) to replace the company as an allocation system, thus moving from an authority-based system (company) to a consensus-based system (market). These classic concepts can be an important reference point at work.

Expanding the market and reducing commercial costs allows to take better advantage of idle resources, reducing slack resources and increasing the rate of generating value (renting the house when you are not, sharing the vehicle when you don't need it, ...). These two aspects seem to favor the lower layers of society (the idea that is transmitted at work through “democratization”), improving “social justice”, ... (this must be well-argued).

The work sows doubt about who will appropriate the value generated (platforms or consumers) through various references, this interesting issue remains unresolved throughout the work. This issue is very important in order to be able to establish international legislation for sharing platforms, which may be using international trade law as a reference and adapting it as much as possible.

The research of the literature must be more precise, critical and profound, that is, the concepts must appear at the beginning of the text, their definition must be precise and their analysis must be well structured.

Here are some suggestions following the discursive structure of the paper.

Introduction

The introduction should be more general and introductory. Perhaps an overview of the “sharing economy” and what aspects have been previously addressed in the literature and what are the main contributions of the work (including here a summary of the methodology and its objectives). The concepts “empower”, “trust”, “social cohesion”, “democratic cohesion”, ... should be discussed at the beginning but in detail, so from my point of view they should be located in a later section. They are concepts that arise a lot at work and are not grouped (the reader is lost when he wants to know that each concept has been previously said). On the other hand, I think that a good definition with a clear border for each concept, avoiding intersections, would add value to the work.

The level of bibliographic review does not make it necessary to establish a different section of the introduction. The important thing in this part is to establish initial general references that raise the subsequent argumentation. If the contribution is the clarification of concepts and the discussion of the workshop, in order to establish indicators, it should be aroused here that these concepts are used fuzzily in literature but are important for management.

From my point of view lines, 101-106 can be the basis of the introduction. Line 109 I find hedonic motivations hard to justify.

Social impacts of sharing platforms

This section presents the justification of the concepts used in the search for the best collective management. This must be clear but it is difficult work. From my point of view the emergence of “sharing platforms” are a cause, not a consequence of social cohesion, increased confidence, ... although all this has to be justified well (line 126). Aspects such as inclusion or exclusion are in the air.

The idea that “sharing platforms” link performance to skills and less to scale economies, and this is an aspect sociality integrative should be highlighted and developed. In this sense, education plays a key role in avoiding exclusion.

“Assessing social sustainability” must be included in the previous point.

Methodology

This section should serve to clarify and specify the concepts already presented in the previous section, justifying the choice made according to the methodology. Some technique, a little more sophisticated, is missed to make ranking (for example AHP). New concepts should not appear here. Once the concepts have been chosen, the indicators for each concept must be justified. The number of concepts and indicators should be reduced and concreted.

Participation in governance must be clarified, which means and how it is done. Getting feedback from users has always been done, but that doesn’t mean they’re involved in governance.

It tries to cover a lot with little justification, such as talking about personal growth.

Problems arise from the limits of concepts, is the same “empowers” as “Stakeholder collaboration” or “inclusivity”? What are the differences?

Finally, the drafting should be reviewed, as an example, line 96 "We use the term user to include the actors involved...". Line 428 “ICT” has not been previously defined.

Reviewer #2: Thank you for the opportunity to read and comment on this submission. This paper first analyzes and describes the social impacts indicators of sharing platforms according to the existing literature, and then the key part of the paper is the opinions and analysis of the stakeholder seminar. Then, the results of the stakeholder seminar and the content of the literature are integrated, and 18 indicators included in four social aspects are determined to form the evaluation framework. All of them are narrated and have a literature basis. However, the following issues may be solved further.

The focus of this paper is to evaluate the social impacts of sharing platforms. The paper summarizes the indicators describing the social impacts of sharing platforms in terms of social sustainability. Why to study the indicators of social sustainability, the paper needs to explain the relationship between social impacts of sharing platforms and social sustainability, or point out the definition of social impacts of sharing platforms in this paper.

This paper classifies the stakeholders, but I still can't clearly see the impacts of sharing platforms on them or the relationship between them, whether positive or negative. I hope that this description may be added.

Resource users and resource owners are important participants in sharing economy. Citizens are the important group in the evaluation of social impact. Why these three groups are not included in workshop?

According to the stakeholder's ranking of the four social aspects, the importance, grade or weight of the indicators has obtained certain results. Then in the fourth step of the paper structure, it is proposed to carry out data analysis, because the paper aims to provide measurable indicators to evaluate the social impacts of sharing platforms, the paper does not give the specific data analysis process, also do not enumerate the measurement results of the relevant data analysis, adding data analysis can show the rationality and scientificity of the system framework.

This study proposes a framework to assess social impact of sharing platforms. Why did it assess from four dimensions (platform, resource owner, resource user, and the society). Can it represent the boundary of social impact?

The last point is the relationship between the cultural background and the indicators proposed in this paper. Whether the indicators proposed in the paper are applicable to the social impacts of sharing platforms in all cultural environments remains to be discussed. If the premise description of social and cultural background can be given, this paper will make greater contributions.

Reviewer #3: In my opinion the manusript is technically correct and comply with journal guidelines. The problem of sharing platform is new one and may be the object of multidisciplinary research. I would like to note a view the practice of writing about non-heterosexual human behaviours as a condition of alleged discrimination is not appropriate for the prestige of science, furthermore for the moral values and common good.

6. PLOS authors have the option to publish the peer review history of their article (what does this mean?). If published, this will include your full peer review and any attached files.

Reviewer #1: No

Reviewer #2: No

Reviewer #3: No

---

## [Author Response · Author response to Decision Letter 0]

14 Sep 2020

• We have followed the PLOS ONE style guidelines, including for file naming

• We have amended the title in the manuscript to match the title in the online submission form

• We have verified that we have captions for our Supporting Information Files at the end of our manuscript, which correspond to the in-text citations 

• All of our underlying data is available as supporting information

See corresponding file for detailed responses to each reviewer.

---

## [Editor Report · Decision Letter 1]

25 Sep 2020

Systematic framework to assess social impacts of sharing platforms: Synthesising literature and stakeholder perspectives to arrive at a framework and practice-oriented tool

PONE-D-20-15696R1

Dear Authors,

We’re pleased to inform you that your manuscript has been judged scientifically suitable for publication and will be formally accepted for publication once it meets all outstanding technical requirements.

Kind regards,

Dejan Dragan, PhD

Academic Editor

PLOS ONE

Additional Editor Comments (optional):

The AE has investigated the corrected paper. All major issues have been appropriately corrected, and comments have been adequately followed. Moreover, all the Reviewers’ questions and dilemmas have been appropriately explained or revised. The paper obviously brings certain novelties, is written in the intelligent and professional style, and represents contributions to the research field. Accordingly, the AE believes that the paper might have been worth to be accepted and placed in the further publishing process.

Academic Editor DD
---

## [Editor Report · Acceptance letter]

28 Sep 2020

PONE-D-20-15696R1 

Systematic framework to assess social impacts of sharing platforms: Synthesising literature and stakeholder perspectives to arrive at a framework and practice-oriented tool 

Dear Dr. Curtis:

I'm pleased to inform you that your manuscript has been deemed suitable for publication in PLOS ONE. Congratulations! Your manuscript is now with our production department. 

Kind regards, 

on behalf of

Dr. Dejan Dragan 

Academic Editor

PLOS ONE